# Downstream high-speed plasma jet generation as a direct consequence of shock reformation

Savvas Raptis [1✉], Tomas Karlsson[1], Andris Vaivads [1], Craig Pollock[2], Ferdinand Plaschke [3,4], Andreas Johlander[5,6], Henriette Trollvik[1] & Per-Arne Lindqvist [1]

Shocks are one of nature's most powerful particle accelerators and have been connected to relativistic electron acceleration and cosmic rays. Upstream shock observations include wave generation, wave-particle interactions and magnetic compressive structures, while at the shock and downstream, particle acceleration, magnetic reconnection and plasma jets can be observed. Here, using Magnetospheric Multiscale (MMS) we show in-situ evidence of high-speed downstream flows (jets) generated at the Earth's bow shock as a direct consequence of shock reformation. Jets are observed downstream due to a combined effect of upstream plasma wave evolution and an ongoing reformation cycle of the bow shock. This generation process can also be applicable to planetary and astrophysical plasmas where collisionless shocks are commonly found.

[1] Division of Space and Plasma Physics - KTH Royal Institute of Technology, Stockholm, Sweden. [2] Denali Scientific, Fairbanks, AK 99709, USA. [3] Institute of Geophysics and Extraterrestrial Physics, Technische Universität Braunschweig, Brunswick, Germany. [4] Space Research Institute, Austrian Academy of Sciences, Graz, Austria. [5] Department of Physics, University of Helsinki, Helsinki, Finland. [6] Swedish Institute of Space Physics, Uppsala, Sweden. ✉email: savvra@kth.se

Earth's bow shock, resulting from the interaction of the super-magnetosonic solar wind and Earth's magnetic field, has been studied for over 50 years and due to the availability of in-situ measurements, serves as an ideal astrophysical laboratory to study collisionless shocks[1–3]. The type of bow shock that is most challenging to study is the so called quasi-parallel shock, where the upstream magnetic field is approximately parallel to the shock's surface normal[4,5]. Downstream of it, the shocked solar wind forms a highly variable environment named the magnetosheath. The shock and its upstream and downstream region create a complex environment in which several magnetospheric phenomena of diverse nature have been observed, like Short Large Amplitude Magnetic Structures (SLAMS), reconnecting current sheets, and fast plasma flows[4,6,7]. The quasi-parallel shock itself is a place that is dynamically evolving, giving rise to several phenomena embedded in its structure. It has been shown that the quasi-parallel shock contains local curvature variations (ripples)[8–11]. Furthermore, the shock is dynamically evolving through its interaction with the foreshock waves upstream of it. These waves evolve, and get steepened to a larger amplitude as the solar wind brings them back to the shock. Their interaction with the shock environment gives rise to a new shock front, while the previous one convects into the magnetosheath region (reformation)[5,10,12–16].

One important property of quasi-parallel shocks is the formation of downstream jets with high dynamic pressure, well above the solar wind dynamic pressure[9,17,18]. They have been suggested to trigger magnetopause reconnection[19], excite surface eigenmodes on the magnetopause[20] and accelerate electrons[21]. Some proposed generation mechanisms connect jets to the solar wind interaction with the local inclination of bow shock ripples[9,18,22,23] or to solar wind discontinuities[24]. Although several mechanisms have been proposed to explain how jets are generated, their origin is still not understood. Some studies have speculated on the connection of jets to upstream magnetic compressive structures (e.g., SLAMS)[25–27]. However, no direct observations have been made so far and the exact causal link has yet to be revealed.

In this work, we use data from recently available unique string-of-pearls configuration of the four Magnetosphere Multiscale MMS spacecraft[28] that allow to follow the jet formation at the shock. In contrast to earlier suggested mechanisms, we show that high-speed jets downstream of the quasi-parallel bow shock can be generated as a direct consequence of the upstream wave evolution and the bow shock reformation cycle. Furthermore, we observe localized downstream density enhancements (embedded plasmoids[23,25]) generated by the same process. The string-of-pearls configuration and the relatively stable shock conditions allow us to observe the development of both phenomena, originating at the upstream region, evolving and ending up downstream in the magnetosheath.

## Results

**Observational overview.** We use data from the MMS spacecraft[28] on 2019-02-12 from 14:56:50 UTC to 14:58:20 UTC. Figure 1 shows the satellite separation in the $xy$ and $xy$ plane, which are effectively identical (string-of-pearls configuration). Figure 2a, b provides ion and magnetic field measurements for MMS2 and MMS1, during the corresponding period while Fig. 3a, b provides ion and magnetic field measurements for MMS2 and MMS1. Figure 4 provides detailed measurements, during the jet observations, for the outermost (MMS2, MMS1) spacecraft. Furthermore, Fig. 5 shows 2D reduced velocity distribution functions (VDFs) for MMS2 and MMS3, while Fig. 6 provides jet-associated measurements for the innermost (MMS4, MMS3) satellites. Starting at 14:57:06, MMS2 observes a localized

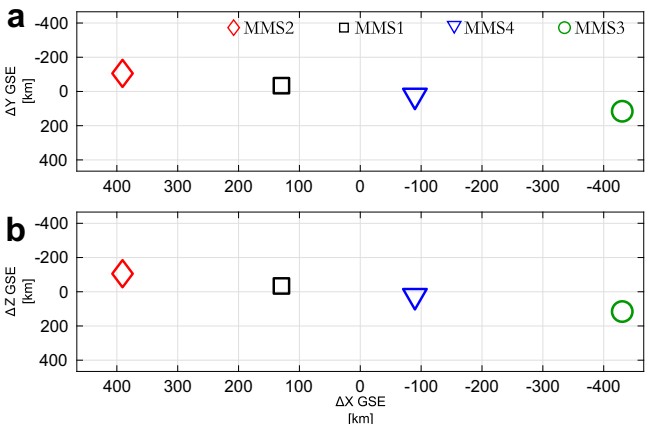

**Fig. 1 Spacecraft formation for Magnetosphere Multiscale (MMS) 1–4.** Spacecraft separation in the **a** $xy$ plane and **b** $xz$ plane in Geocentric Solar Ecliptic (GSE) coordinates. Note that the string-of-pearl configuration effectively provides an identical formation in the two planes.

structure of increased magnetic field and density (red shaded region, 1). The magnetic structure is elliptically polarized (left-hand in the spacecraft reference frame) and as discussed below is traveling towards the Earth. These properties along with the scale size being ~1000 km and the localized increase in |**B**| and density correspond to typical properties of a SLAMS ([8,29,30]). However, in order to properly classify each structure as SLAMS, one needs to evaluate whether it satisfies a set of criteria (e.g., see[30]). As this is out of the scope of this work, we will use the term compressive magnetic structure. As observed by MMS2, this structure is initially upstream from the Earth's bow shock, while from the point of view of MMS1, and MMS4 is effectively the local bow shock outer edge. Upstream of it, we observe a region of waves called whistler precursors (blue shaded region) that are typically observed upstream of SLAMS. These whistler waves have been linked to shock reformation dynamics[1,2,31]. A few seconds later, as shown in panels Figs. 3a and 6a, MMS4 observes another structure emerging (region 2) between 14:57:38 and 14:57:47. This structure is generating another shock transition, spatially and temporally detached from the first one (region 1). Finally, another transition from the upstream to downstream is observed at approximately 14:57:50. This transition is associated with another localized density and magnetic field enhancement region (red shaded region 3) observed by MMS2 at approximately 14:58:00.

**Time evolution and environment characterization.** The overview observations shown in Figs. 2, 3 and the detailed observations of Figs. 4, 6 along with the 2D VDFs (Fig. 5d) show that MMS3 is situated downstream of the bow shock, during the whole time interval (|**B**| and $n$ are significantly higher than the local solar wind measurements, and the VDFs show a thermalized ion population). The other satellites, hundreds of kilometers away, reside upstream, observing the corresponding solar wind and foreshock regions. A direct view of the evolution of the initial shock (region 1) can be seen in Fig. 7. There, by using MMS1 as reference, the magnetic field measurements of MMS2-4 have been time-shifted by cross-correlating all spacecraft observations (see methods Cross-correlation and timing). This effectively allows us to view the evolution of the initial shock (region 1) and its corresponding upstream waves connected to the magnetosheath jet observed by MMS3. The dynamical evolution of the shock's ramp (patterned red region) is first visible in MMS1 but is more prominent in MMS4. This evolution is consistent with previous

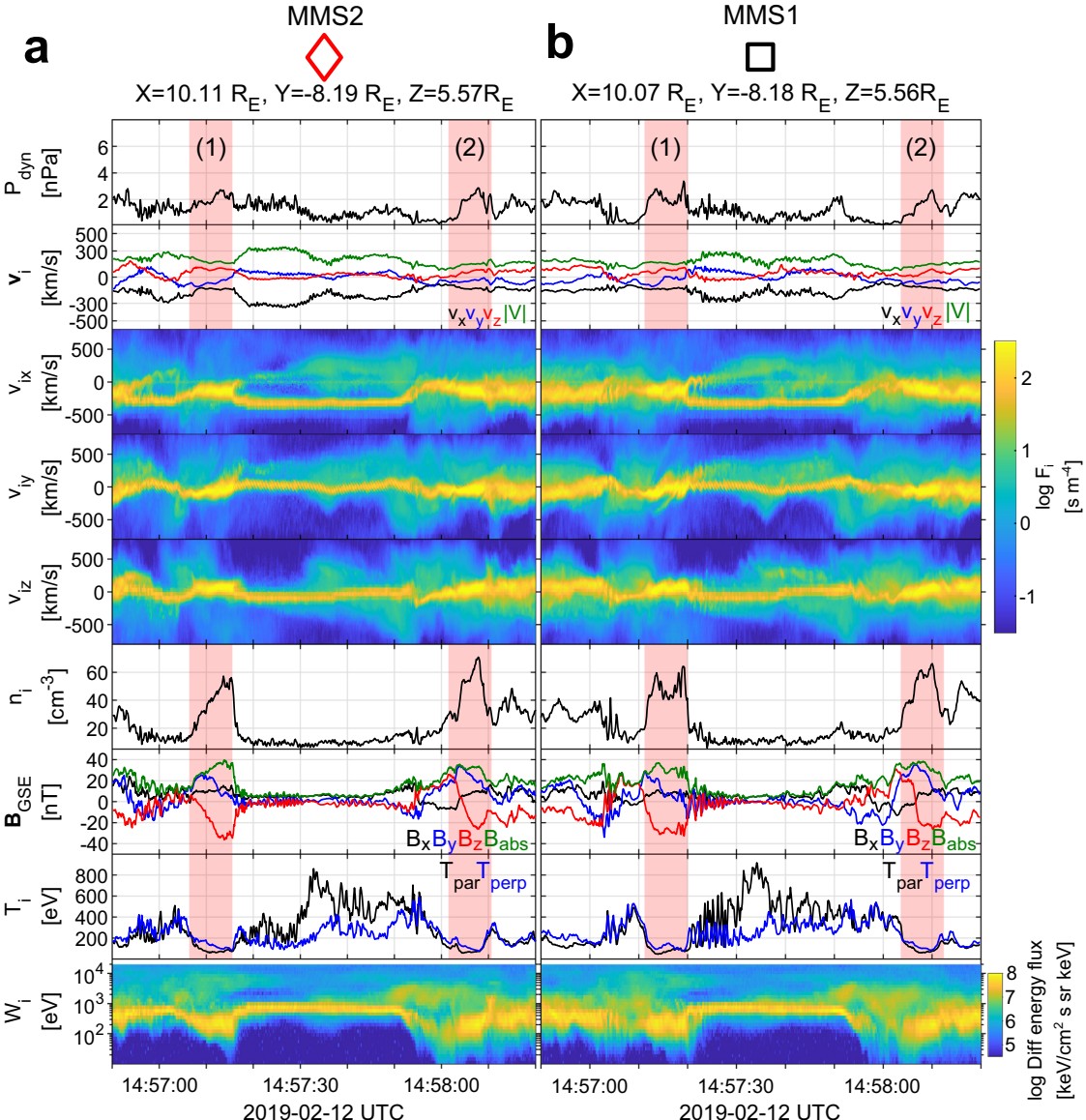

**Fig. 2 Spacecraft position and overview of the measurements for Magnetosphere Multiscale (MMS) 2 and 1. a, b** Overview observations in Geocentric Solar Ecliptic (GSE) coordinates of the 2 outermost MMS spacecraft ordered by distance to the Earth (MMS2 - red diamond, MMS1 - black square). From top to bottom, ion dynamic pressure, ion bulk velocity flow, $v_x$, $v_y$, $v_z$ 1D reduced ion velocity distribution functions (VDFs), ion number density, magnetic field, ion temperature, and ion differential energy flux spectrum. Sequentially observed shock fronts are numbered and marked with red-shaded color.

computer simulations[32,33] and other observational studies[31,34]. Moving downstream of the shock, as observed by MMS3, the evolved structure (region 1) appears as a shock remnant of relatively enhanced density and magnetic field located in the magnetosheath region. The structure now, as viewed in the magnetosheath, includes a pile-up region (patterned red region) of the waves associated to the shock's ramp evolution. Similar events have been discussed in recent studies[23,35]

At approximately 14:57:35–14:57:45 (as viewed in Fig. 6a), MMS4 observes a new compressed plasma region (region 2), spatially detached from the initial shock (region 1), forming upstream of the first, becoming the new local shock front, and thus completing a bow shock reformation process/cycle. Returning to the global picture shown in Figs. 2, 3, one can now note that this reformation process arises again with the appearance of region 3. As a result, locally, the outer edge of the shock change, following the numbered shaded regions 1, 2, and finally 3.

This process can explain the different observations made by the two outer spacecraft (MMS 1–2) and the inner ones (MMS 3–4). This process has been hypothesized and reported in simulations of the quasi-parallel shock[5,12,13,15,16].

**Jet observations**. Having established the different regions and the shock reformation process, we proceed to interpret the super-magnetosonic jet observations made by MMS3. For this observation, an explanation is required for the enhanced bulk ion velocity and density of the jet. The full particle moments (Figs. 3b and 6b) show that inside the jet $|\mathbf{v}| \approx 220$ km/s and $n \sim 60$ cm$^{-3}$. The jet however contains two different ion populations, a background magnetosheath, and a beam-like jet. Calculating the moments for the beam-like part of the distribution, we obtain $|\mathbf{v}| \sim 350$ km/s and $n \sim 40$ cm$^{-3}$ (see Figs. 5d and 6c). This corresponds to a relative increase of ~200% in dynamic pressure

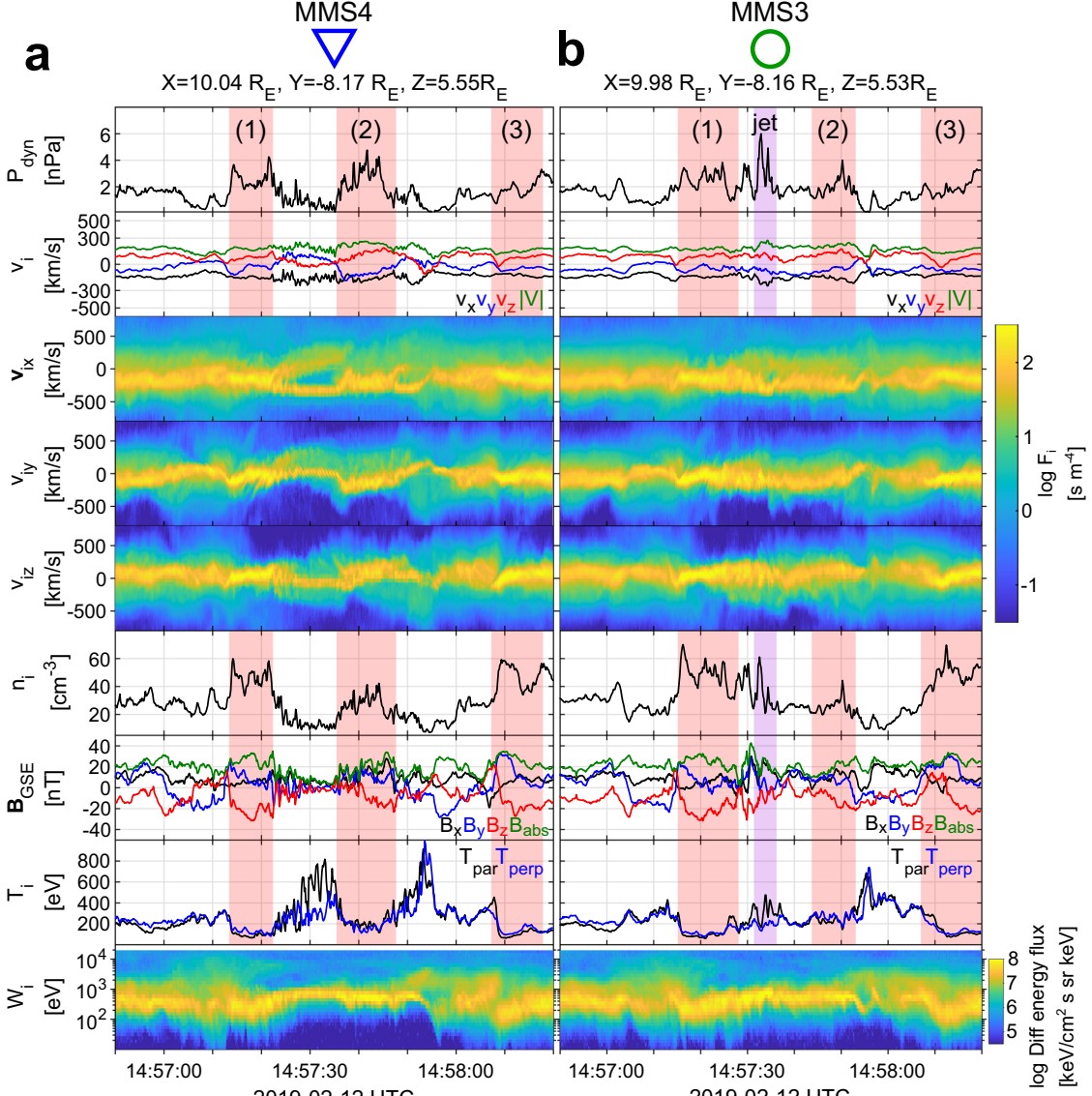

**Fig. 3 Spacecraft position and overview of the measurements for Magnetosphere Multiscale (MMS) 4 and 3. a**, **b** Overview observations in Geocentric Solar Ecliptic (GSE) coordinates of the two innermost MMS spacecraft ordered by distance to the Earth (MMS4 - blue triangle, and MMS3 - green circle). From top to bottom, ion dynamic pressure, ion bulk velocity flow, $v_x$, $v_y$, $v_z$ 1D reduced ion velocity distribution functions (VDFs), ion number density, magnetic field, ion temperature, and ion differential energy flux spectrum. Areas of interest have been marked with colors, red for sequentially observed shock fronts (which are also numbered), and purple for the observed jet.

compared to both background magnetosheath and solar wind levels. Thus, the beam-like jet population has higher density but very similar velocity and temperature to the solar wind observed by the other MMS spacecraft upstream of the shock. The observed increase in density appears to be linked to the whistler precursors, similarly to what has been shown in other recent studies (e.g.,[31,36]). Furthermore, the exact observations may be explained by the non-linear evolution of the observed pulsations[31,37] or due to a gyro-trapping mechanism originating from the evolution of the whistler waves upstream of region 1 as recently discussed[34]. Specifically, we observe enhancements in plasma density and magnetic field magnitude, similarly to other observational studies[31]. Finally, the enhanced velocity of the jet relatively to the magnetosheath can be explained by the effect of the reformation cycle. The beam-like jet population, found within the evolving upstream waves, is effectively transferred downstream of the shock, having little to no interaction with the shock environment imposed by the initial compressive structure

(region 1). Through the reformation process, a new shock front forms upstream of the jet, enclosing it with thermalized magnetosheath plasma and thus completing its formation.

**Jet generation and reformation process.** Combining all the observations above, we infer the sketch in Fig. 8 summarizing the jet generation mechanism. The initial shock (region 1) is first observed by MMS2 between 14:57:06 and 14:57:17—time $t_1$ in Fig. 8). This region is initially detached from the Earth's bow shock (Figs. 2, 4) but due to its propagation towards Earth (Fig. 7) it eventually forms the outer edge of the bow shock (MMS1 and MMS4), while finally ending up downstream of the bow shock (MMS3). During this time, region 1 interacts with the faster upstream waves, forming a pile-up region. Thus, when viewed downstream, by adding the pile up region, the whole structure has grown through evolving in time, creating an extended region of increased magnetic field and plasma density. These observations

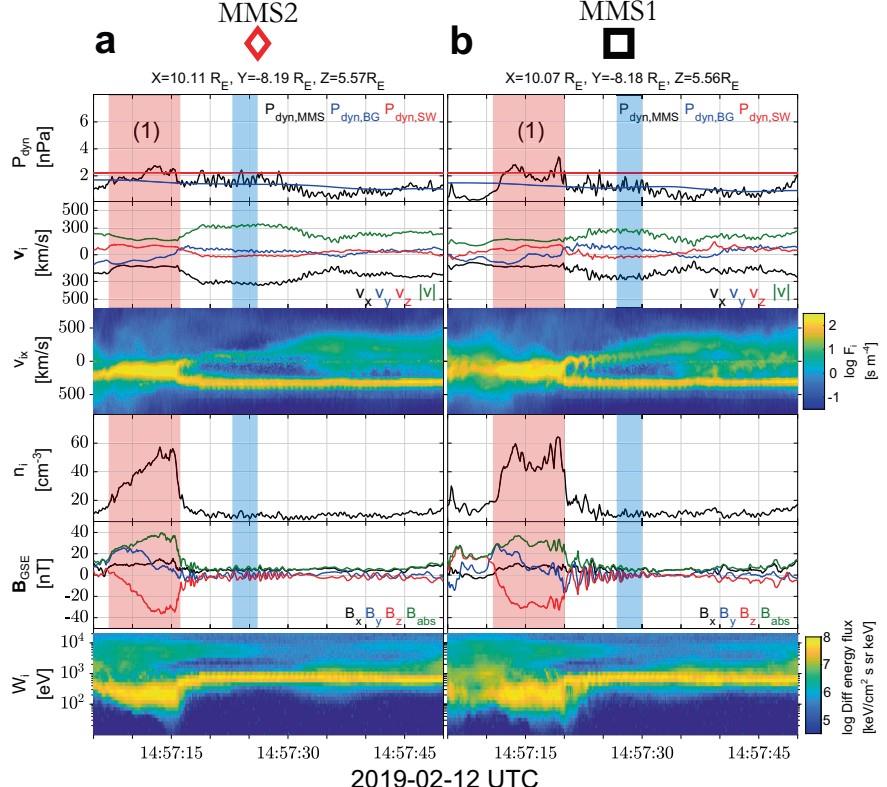

**Fig. 4 Zoomed in view of the measurements for Magnetosphere Multiscale (MMS) 2 and 1. a, b** Detailed observations of the outermost MMS spacecraft 2 and 1, ordered by furthest to closest to the Earth. (Top-bottom): ion dynamic pressure along with solar wind and magnetosheath background level, ion velocity, reduced 1D Velocity Distribution Function (VDF) in the *x*-direction, ion number density, magnetic field measurements, and differential energy spectrum. Special indication has been made regarding the compressive magnetic structure (red region 1), and the corresponding upstream waves (blue region). Shaded regions are approximated using the methodology shown in Fig. 7 and discussed in "Methods", subsection Cross-correlation and timing.

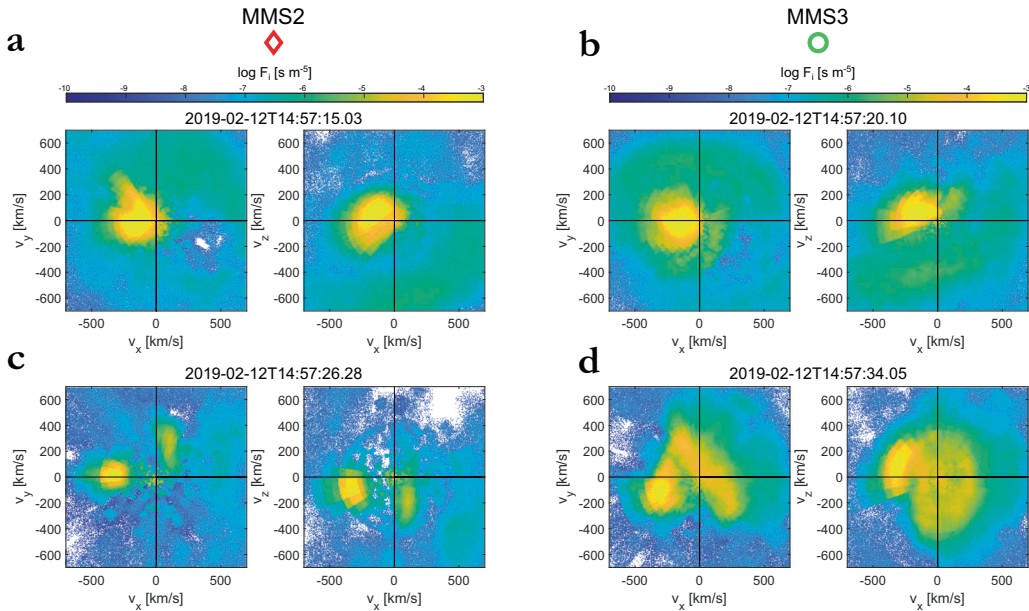

**Fig. 5 2D velocity distribution functions (VDFs) for each observed structure.** Reduced 2D VDFs for **a** the compressive magnetic structure/shock (red shaded region 1 of Figs. 2a and 4a) as observed by the outermost spacecraft, Magnetosphere Multiscale (MMS) 2. **b** The same structure as observed by the innermost spacecraft, MMS3. **c** Upstream waves (blue shaded region 1 of Figs. 2a and 4a) as observed by the outermost spacecraft, MMS2. **d** Magnetosheath jet (purple shaded region 1 of Figs. 3b and 6b, c) as observed by the innermost spacecraft, MMS3. All projections are in Geocentric Solar Ecliptic (GSE) coordinates.

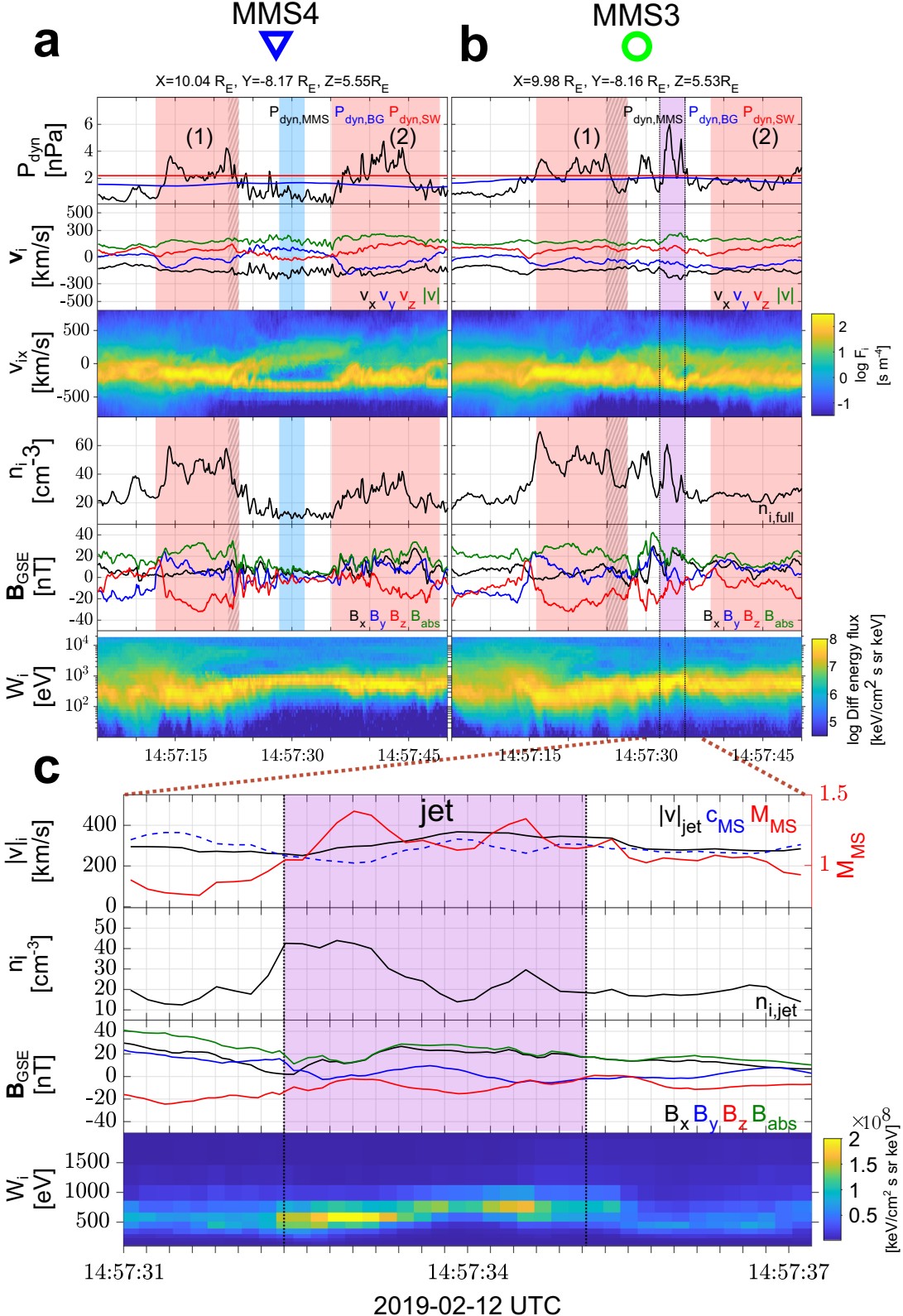

**Fig. 6 Zoomed in view of the measurements for Magnetosphere Multiscale (MMS) 4 and 3. a**, **b** Detailed observations of the innermost MMS spacecraft 4 and 3, ordered by furthest to closest to the Earth. (Top-bottom): ion dynamic pressure along with solar wind and magnetosheath background level, ion velocity, reduced 1D Velocity Distribution Function (VDF) in the *x*-direction, ion number density, magnetic field measurements, and differential energy spectrum. Special indication has been made regarding the compressive magnetic structures (red regions 1 and 2), the pile-up region (patterned red region), the jet observation (purple), and the corresponding upstream waves (blue region). Shaded region are approximated using the methodology illustrated in Fig. 7 and discussed in the methods subsection Cross-correlation and timing. **c** Shows the partial moments (see "Methods" subsections Jet definition and MVA and distribution functions.) for the jet observed by MMS3. From top to bottom, jet's bulk velocity, along with fast magnetosonic speed and fast magnetosonic Mach number, ion number density, magnetic field components, and differential ion energy spectrum.

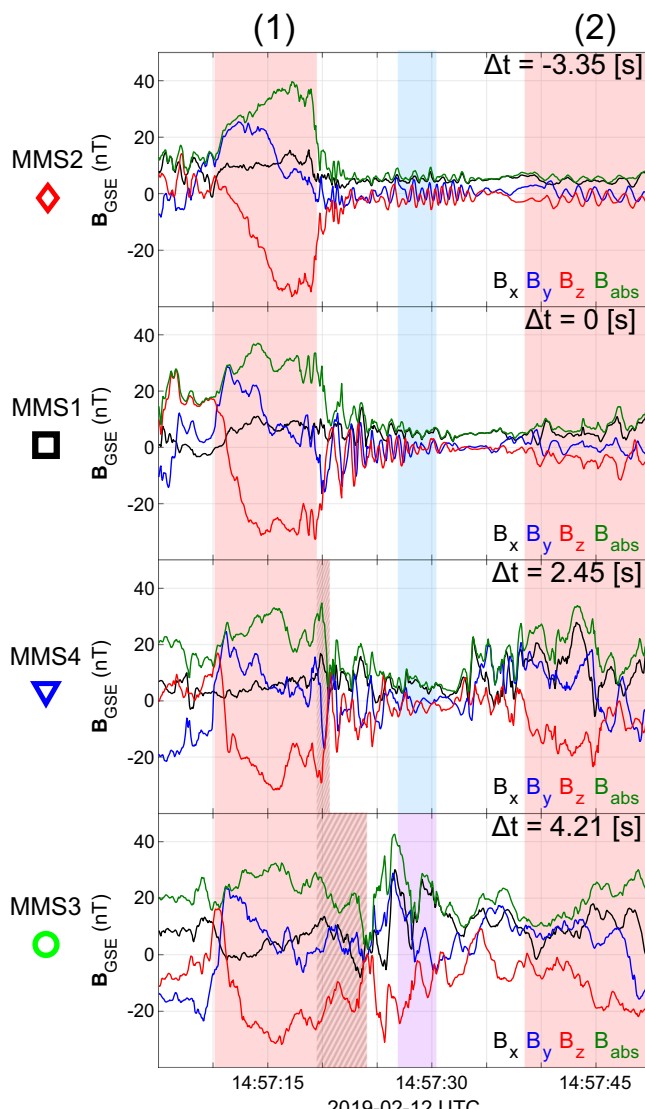

**Fig. 7 Time-shifted magnetic field measurements for Magnetosphere Multiscale (MMS) 1–4.** From top to bottom, magnetic field measurements for MMS2, MMS1, MMS4, and MMS3. All measurements are time-shifted with respect to MMS1 (black square) and each panel indicates the time lag used for each spacecraft. All panels effectively show the evolution of the magnetic field from a point moving with the trailing edge of the (red-shaded region 1). The pile-up region observed by MMS4 and MMS3 is indicated by the patterned red region. This pile-up corresponds to the evolution of the shock's ramp that extends the initial compressive structure. The **B** measurements corresponding to the magnetosheath jet (MMS3) are shown in purple and the corresponding upstream waves observed by MMS1, 2, and 4 are in blue.

correspond to the equivalent downstream phenomenon of embedded plasmoid or shock remnant at the magnetosheath[23,25]. Note that this structure, due to the pile-up process, appears in the magnetosheath as an extended region that includes several structures (i.e., SLAMS and whistler waves). As a result, while the individual parts of the structure may maintain their polarization and overall properties, the whole region becomes hard to distinguish, and its origin would have been unknown in the absence of upstream measurements. A second compressed plasma region, forming upstream of region 1, is then observed by MMS4 at 14:57:40 ($t_4$). This effectively creates a reformation cycle, between

the old shock front (region 1) and the newly formed one (region 2). Finally, the whole event (Figs. 2, 3) is completed by another reformation process caused by the region 3 first observed by MMS1 at ~14:58:00. The described shock transitions are observed sequentially from the spacecraft reference frame, starting from the spacecraft the furthest away from the Earth (MMS2) and eventually reaching the satellite closest to the Earth (MMS3). This allows us to map the evolution of all the phenomena, as illustrated in Fig. 8. From the evolution of the whistler waves upstream of the initial shock (region 1) and the dynamical evolution of the bow shock, a super-magnetosonic downstream jet is generated. The jet can be viewed as a plasma population that due to its minimal interaction with the already weakened shock front (region 1), it retains its solar wind-like properties (beam-like structure). While initially generated at the very edge of the shock, due to the reformation cycle, the jet is effectively transferred downstream of the bow shock while its bulk velocity shows it is directed towards the Earth (Fig. 6b, c).

## Discussion

We have observed signatures of localized compressive magnetic structures (i.e., SLAMS/Shocklets) forming the local bow shock front and evolving until the end of their lifetime, when they dissolve into becoming the downstream magnetosheath. More importantly, we have shown direct observations of downstream super-magnetosonic jets generated directly from the evolution of upstream waves and the shock reformation cycle, thus suggesting a mechanism for the jet formation. This mechanism is fundamentally different from the previously proposed ones, which require the presence of external factors (e.g., discontinuities[24]) or specific geometric configurations (e.g., ripples[22]) to take place to explain jets' generation. In the presented model, the downstream jet phenomenon is generated as a natural part of the dynamical evolution of collisionless shocks. These results are not only important to near-Earth space but also to planetary and astrophysical plasmas where collisionless shocks are ubiquitous[2,38–40]. The results of this work are a direct success of the MMS mission that with its state-of-the-art instruments and the string-of-pearls configuration has enabled the discovery of the jet formation mechanism. Specifically, this discovery is of high importance for designing and operating future spacecraft missions studying the bow shock. Further work could address in detail the exact properties of the ion trapping mechanism and the corresponding role of electron dynamics. More importantly, it is still unknown if similar mechanisms can produce more extended flows downstream of the bow shock. Furthermore, the exact properties and characterization of the observed compressive magnetic structures (numbered 1–3) is another vital continuation of this work to provide a full modeling of the dynamical evolution of the bow shock. Finally, a statistical analysis and a comparison with global simulations is the next logical step.

## Methods

**Data.** In this work, data from the MMS mission[28] are primarily used, while for the estimation of upstream solar wind dynamic pressure, we use the OMNIWeb database[41]. The magnetic field measurements of MMS are from the fluxgate magnetometer (FGM)[42] of the FIELDS instrument suite[43], sampled every 0.0625 s. The fast plasma investigation (FPI) instrument[44] gives distribution function measurements with a cadence 0.15 s for ions and 0.03 s for electrons. The solar wind dynamic pressure and the model location of the bow shock and magnetopause positions are taken from the OMNIWeb dataset[41]. All the vector quantities are in Geocentric Solar Ecliptic (GSE) coordinates.

**Jet definition.** For Figs. 2, 3 and Figs. 4, 6, the dynamic pressure is calculated as: $P_{i,dyn} = \rho_i \cdot |\mathbf{v}|_i^2 = n_i \cdot m_p \cdot |\mathbf{v}|_i^2$. The background magnetosheath dynamic pressure is calculated by using a moving average window of 30 s. These are used for defining and classifying the jet observations for which we use the same definition as other related works (e.g.,[25,26,45,46]). In particular, we define a jet as the time interval

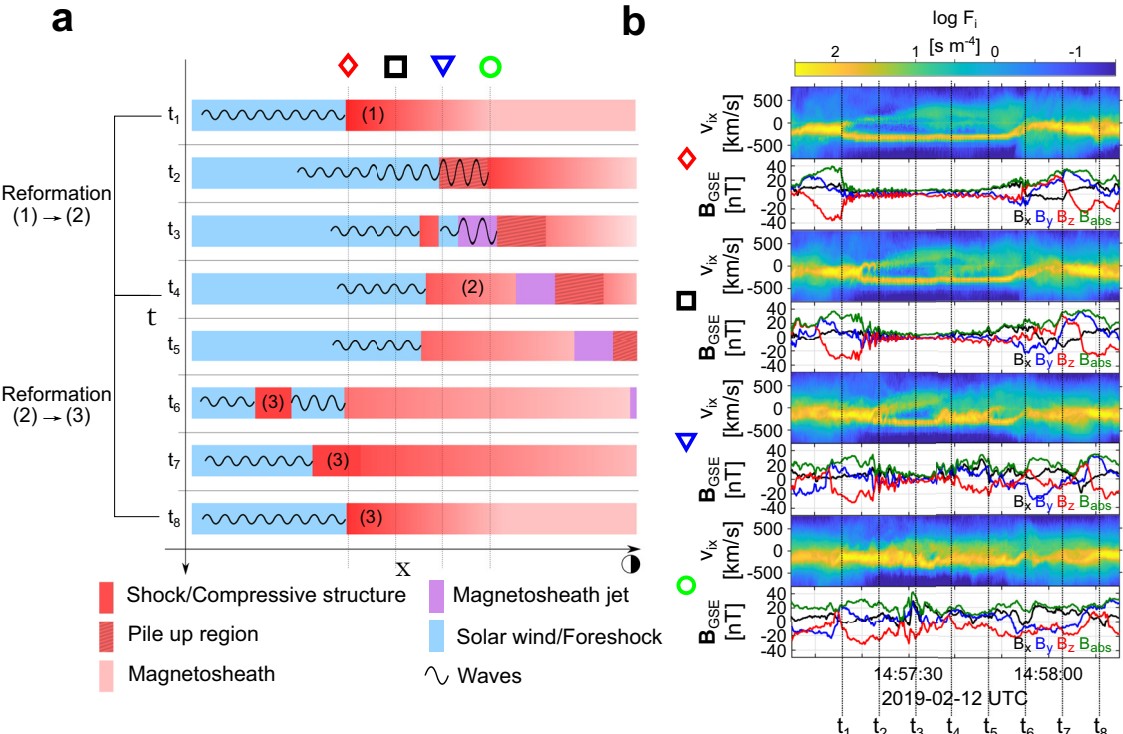

**Fig. 8 Magnetosheath jet formation mechanism. a** Sketch of the jet formation mechanism as a result of the bow shock reformation cycle. Magnetosheath jet appears as compressed solar wind that is effectively observed downstream of the shock through the generation of a secondary local shock front, upstream of the initial one. Note that due to the string-of-pearl formation (see Fig. 1) the satellites are aligned in the x Geocentric Solar Ecliptic (GSE) coordinate **b** reduced 1D velocity distribution functions (VDFs) and magnetic field measurements per spacecraft going from the furthest away from the Earth to the closest. The interpretation of the satellite measurements (Figs. 2–6) are indicated by the vertical dotted lines. The numbering of the different magnetic structures corresponds to the same structures as shown in the above figures. All the information that does not appear in the measurements (vertical lines of panel **a**) are inferred from the evolution of the observed structures and therefore are speculative.

where the dynamic pressure is at least 100% higher than the background value (see Figs. 4, 6 panels a, b, blue line). Specifically, the observed jet exhibits an increase of ~200% compared to both the magnetosheath and the corresponding solar wind dynamic pressure, which is well above the typical threshold level. For the jet definition, we use plasma moments derived from the FPI instrument of MMS. It should be noted that the plasma moment derivation, in close to the solar wind environments, can contain statistical uncertainties (e.g.,[47]) as well as physical uncertainties from the particle detector. However, in our case these uncertainties do not provide any issue with the definition or the characterization of the jet since the observations are well above the threshold we use.

**Characteristic velocities**. For Fig. 6c, the sound speed is estimated as $c_s = \sqrt{(KT_e + \gamma_i KT_i)/M}$, where $K$ is the Boltzmann constant, $T_{i,e}$ is the ion and electron temperature, $M$ is the proton mass, and $\gamma_i = 3$. The Alfvén velocity is calculated as $c_A = |\mathbf{B}|/\sqrt{\mu_0 \cdot \rho_i}$, where $\rho_i$ is the ion mass density, and the magnetosonic velocity is computed as $c_{MS} = \sqrt{c_s^2 + c_A^2}$. Futhermore, the equivalent fast magnetosonic Mach number ($M_{MS} = |\mathbf{v}|_i/c_{MS}$) is derived and ploted in Fig. 6. any Mach numbers above unity correspond to super-magnetosonic observations.

**Cross-correlation and timing**. For Fig. 7, MMS1 is used as a reference spacecraft to time-shift the measurements of the other spacecraft so that the time-series represent a co-moving view from the initial shock (region 1). The required time shift is obtained from the maximum sample cross-correlation peak lag[48]. The maximum correlations between the signals for the time lags used are $\rho_{1,2} = 0.87$, $\rho_{3,2} = 0.44$, $\rho_{4,2} = 0.78$. More details can be found in chapters 12 and 13 of[49].

**MVA and distribution functions**. The polarization of the compressive magnetic structure (region 1) is obtained with the help of the minimum-variance analysis (MVA) (see chapter 8 of[49] and chapter 1 of[50]. For the reduction of ion distribution to generate the 2D projections (Fig. 5), 3 sequential measurement points from the FPI instrument were used. Partial ion moments of the jet-like population of the magnetosheath jet are computed based on the 2D reduced distribution functions, where the specific ion population can be clearly seen at negative $v_x$ and $v_y$ as shown in Fig. 5.

## Data availability

Satellite mission data analyzed in this study are publicly available via the repositories of each satellite mission. Magnetospheric Multiscale (MMS) measurements are available through https://lasp.colorado.edu/mms/sdc/public/about/browse-wrapper/or through the Graphical User Interface (GUI) found in https://lasp.colorado.edu/mms/sdc/public/search/. The OMNI high-resolution data are available through https://omniweb.gsfc.nasa.gov/form/omni_min.html. The data used in this study are also available in the associated GitHub database, https://github.com/SavvasRaptis/Jets-Reformation[51]. The datasets generated during and/or analyzed during the current study are available from the corresponding author on reasonable request.

## Code availability

The implementation of the minimum-variance analysis, the reduction of the VDFs to 2D and 1D, and the computation of the partial moments based on specific parts of the reduced 2D VDFs is done via the functions of irfu-matlab package, openly available at https://github.com/irfu. Examples and documentation for each of the function may also be found directly in the irfu-matlab package repository page at https://github.com/irfu. The cross-correlation and timing analysis is performed using the sample cross-correlation function as implemented in MATLAB software, https://mathworks.com/help/econ/crosscorr.html

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

## Acknowledgements

We thank the MMS team for providing data and support. We acknowledge the use of NASA/GSFC's Space Physics Data Facility's OMNIWeb service, and OMNI data. We acknowledge the use of irfu-matlab package, https://github.com/irfu. We thank M. Lindberg and A. Lalti, for their comments on the initial stage of the work. We are also thankful for the useful discussions done with the International Space Sciences Institute (ISSI) team, "Foreshocks Across The Heliosphere: System Specific Or Universal Physical Processes?" S.R. and T.K. are supported by the Swedish National Space Agency (SNSA) grant 90/17. H.T. and T.K. are supported by the SNSA grant 190/19. A.V. was supported by the Swedish Research Council (grant 2018-05514). F.P. is supported by the Austrian Science Fund (FWF): P 33285-N

## Author contributions

S.R. performed the data analysis and wrote the manuscript. T.K and A.V. supervised the study and contributed to parts of the manuscript through reviews and edits. C.P., F.P., A.J., H.T., and P.-A.L. contributed to the writing of the manuscripts through reviews and edits. All authors contributed to the interpretation and the discussion of the results.

## Funding

## Competing interests

The authors declare no competing interests.
