## [Peer Review File · Nature Communications]

REVIEWER COMMENTS

Reviewer #1 (Remarks to the Author):

The authors analyse data of a nicely spaced aligned configuration of the 4 MMS spacecraft covering the range of roughly 1000 km from upstream solar wind across the quasi-parallel bow shock to the downstream near shock magnetosheath. Roughly two spacecraft are in the highly variable upstream wave/shock region, and one shock downstream in the enhanced magnetic field and density magnetosheath. The upstream data show the typical signs of large amplitude upstream waves which are on the way to become the shock by steepening and reflecting particles back upstream which excite precursor whistlers. In order to get a causally connected evolution the spacecraft data have been Galilei shifted in time (this is useful when referring to observations in the magnetosheath where a so-called jet is detected). This evolution of the shock shows the typical signature of shock transitions noting that the front of the 3 shock crossings approaches the spacecraft and passes them over.

The magnetosheath spacecraft shows a structure different from the shock and normal sheath which is identified as the "jet" as a region of enhanced dynamical pressure, consisting of background sheath plasma and a stream-like ion population of nearly solar wind speed and temperature but higher density, i.e. compression. The required anticorrelation between magnetic field and density confirms this picture. The near solar wind speed of the "jet" indicates that it is a structure of the reformed quasi-parallel shock that has passed the shock, survived locally and moves downstream through the sheath. In this picture which the authors develop a sheath is a fraction of an upstream large amplitude foreshock wave packet which finds a dynamical hole to pass barely affected by the shock transition through the shock to the downstream region.

In looking at the solar wind speed (MMS1/2) it seems that the conditions were normal in the solar wind rather than the slower wind speed.

Altogether this carefully done work on the sequence of events related to quasi-parallel collisionless shocks, and the detection of a process of how foreshock features can sometimes be transferred downstream as if no shock would have been present locally is interesting and worth publishing.

I find the observations and the sequence intriguing and a nice contribution to the shock dynamics. It is clear that locally such a jet should affect the evolution of the sheath. The authors present a

graphical model which represents their imagination of how the jet formation proceeds. This may or may not be the ultimate picture. To my taste Fig 4 a is not very helpful as the data already contain the physics in rather clearer form and the model does not clarify it in a better way. However this is a question of taste which may differ from mine.

The observation and identification of the jet and its division into the two populations and relation to shock reformation is most interesting. Obviously in the jet part the reformation has not been successful as it enabled part of the solar wind to pass across the shock, something which has already long been suspected that the shock ramp naturally contains holes where compressed and modified solar wind may pass through on some short tangential scale which is probably prescribed by the tangential wavelength of the large amplitude foreshock waves. Two adjacent of them definitely leave a region where the reformed shock is barely precisely defined.

The investigation and model is uni-dimensional prescribed by the configuration of the quartett of spacecraft. So nothing can be said about the interesting question of the tangential extension of the jet. It must have been tangentially local not distributed allover the surface of the bow shock. And during further propagation it should evolve, either steepening further into another local subshock or decaying by some kind of spatio-temporal dispersion. Magnetosheath flow should transport it sideways. When surviving up to the magnetopause its effect should locally cause a strong deformation of the latter.

Altogether a nice work on which I do not have much criticism to add.

Reviewer #2 (Remarks to the Author):

Summary:

=====

The paper argues that the four MMS spacecraft observed the bow shock reforming and generating supersonic magnetosheath jets in the process. While I fully agree that the evidence supports the interpretation of shock reformation, I am not sure about the observation of a magnetosheath jet. The supplemental figure does look to have a solar wind beam and some much hotter, more tenuous ions drifting relative to it. However, it is not clear this is indeed a jet and that the VDF was even observed downstream of the shock.

As a note, I will say that I am not opposed to the authors' interpretation nor am I opposed to seeing this manuscript again after revision. If the authors are correct in their interpretation, then this certainly would be an incredibly important paper for the physics community at large, not just the heliospheric and astrophysical communities. My main concerns are likely due to my misunderstanding, so I would like to see the authors responses before making any final decisions.

General Comments:

-- Figure 1:

- The label for panel (e) is missing.
- I am not convinced that the purple shaded region is a jet. The reason I say as much is for the following reasons:
 - The FPI DIS instrument is not really suited for measuring the solar wind ions [e.g., Pollock et al., 2016] and as the spacecraft rotates, and DIS scans, the instrument shows a periodicity in the intensity of the ion signal from the main solar wind core beam. If you examine the ion differential energy flux spectrogram and then the 1D reduced ion VDFs, you can see some of this periodicity. But more importantly, you can see that the main beam doesn't really change in the ion differential energy flux spectrogram during what the authors are calling a jet. Perhaps the color scale is saturated or I need new glasses, but I was under the impression that magnetosheath jets show a shift in energy of the peak and an enhancement in intensity of that shifted peak, relative to the surrounding plasma. There is a kind of one-sided shift in the location of the peak but it isn't very convincing. Figure 2 does a little better but I would suggest the authors find a way to make the contrast more obvious because I am really squinting to try and see this.

-- Figure 2:

- The label for panels (d) and (e) are missing.
- Similar to my comment on Figure 1, could you find a way to make the contrast better? Perhaps show the Mach numbers of the partial velocity in panel (e) instead of the relevant speeds? I ask because it's difficult to see just how far above the magnetosonic speed the jet bulk speed lies. This is important as the uncertainty of the velocity moments can be very large even for instruments with much longer integration times. Since FPI

DIS has such a short integration time, the signal-to-noise ratio is very small. This amplifies the uncertainty in the velocity moments. That its energy and angular resolution were designed for the magnetosheath only amplifies the issue in the solar wind. So if this jet is only ~5-10% above Mach 1, is that really statistically significant?

-- Figure 3:

-- The label for MMS4 is missing above the blue triangle.

-- Figure 4:

-- The text label for the purple region is missing.

-- It would be very helpful, for readers like myself, if the MMS symbols also had numbers in this figure. Otherwise folks like myself with poor short-term memories keep scrolling back-and-forth with earlier pages.

-- This figure is difficult to disentangle. On the left, panel (a), the vertical axis is meant to imply time and the horizontal the spatial location along the shock normal, correct? Then I see eight vertical lines in panel (b) that are suppose to correspond to each of the horizontal color-coded bars in panel (a), right? I am confused because at t1 each spacecraft is seeing something very different. I also see the spacecraft symbols above panel (a) and vertical lines extending down. However, panel (a) and (b) doesn't seem to line up. For instance, the red diamond vertical line in panel (a) at t6 looks to be in the upstream solar wind. The vertical line in panel (b) at t6 for the red diamond looks to be in the shock ramp (you can see this from the deceleration of the main solar wind beam in the 1D reduced ion VDFs). Actually all four spacecraft look to be in a shock ramp at t6 according to the location of the t6 line in panel (b). I guess my point is that this figure seems inconsistent or I am completely misunderstanding it.

-- Assuming the purple region corresponds to the jet (inferred because jet is the only type of region missing), then is it the case that only MMS3 sees the jet? I think the authors implied as much in the previous sections, but the 1D reduced ion VDFs here do not really convey or illustrate this, so far as I can tell. I agree it looks like the authors are indeed observing shock reformation in one of its many manifestations, but I am having difficulty discerning why they argue there is a magnetosheath jet here.

SLAMS:

-- Lines 89-93: Note that SLAMS exhibit linear, left-hand, and right-hand polarizations in the spacecraft frame [e.g., Wilson, 2016]. Thus, the left-hand polarization is not really conclusive. The enhancement in $|B|$ simultaneous with n is also not conclusive, as that is just a general property of waves on the fast/magnetosonic-whistler branch. Note that I am not arguing these fluctuations are not SLAMS, just pointing out that the evidence presented is not conclusive.

-- Note that SLAMS are always observed with diffuse ions [e.g., see description and examples in Wilson, 2016] and typically are distinguished from shocklets in that $dB/B > 2$.

-- Only a few of your red-shaded regions actually satisfy this latter criteria and it's not clear if any satisfy the former. The top-left MMS2 (and maybe top-right MMS3) example in Figure S1 kind of looks like diffuse ions, but it's impossible to say without 1D cuts through the VDFs (i.e., not the reduced VDFs). This is because the phase space density profile of diffuse ions is very hard, i.e., it can be nearly flat. My color fidelity is not sufficient to determine the velocity gradient from these 2D reduced

ion VDFs.

-- Even if the top-left MMS2 example in Figure S1 is representative of diffuse ions, is this ion VDF seen simultaneously with what the authors are calling SLAMS? The example red-shaded regions in Figure 1e do not, I think, satisfy the $\text{dB}/B > 2$ criteria.

-- The scale size of SLAMS is only ~ 1000 km along the normal to their direction of propagation and only slightly larger along the transverse [Lucek et al., 2002, 2004, 2008]. The lateral separation between MMS2 and MMS3 is over 800 km, which is coming close to the relevant scale size of SLAMS. So if these oscillations are SLAMS, is it really surprising that you see such large differences in the magnetic field between the different spacecraft? You can see that MMS1 and MMS2 are the most similar in the magnetic field profiles, and these spacecraft are spatially adjacent to each other. The profile starts to degrade at the next adjacent spacecraft, MMS4, then "falls apart" at the furthest spacecraft, MMS3.

-- Lines 172-188:

-- Turner et al. [2021] did not discuss gyrotrapping, so far as I know. They did observe a phase shift between the peak $|dB|$ enhancement and the peak dn enhancement, but this has already been explained as a consequence of nonlinearity [e.g., Stasiewicz et al., 2003].

-- Generally, $|dB|$ and dn oscillate in phase within magnetosonic-whistler precursors [e.g., Chen et al., 2018; Turner et al., 2021], which is what I am seeing in your Figure 2.

-- Lines 198-208:

-- Things that starts upstream and transmit into the downstream are generally compressed, not expanded, correct? Why is the pile-up region expanded in the downstream?

-- How does an oscillation lose its polarization? This would require a mode conversion or some sort of wave-wave interaction or some type of destructive or constructive interference, would it not? That is, you cannot arbitrarily remove angular momentum from the electromagnetic field. So how is it happening here?

=====
Minor Concerns:

=====
-- Lines 14-16: The sentence starting with "The Earth's bow shock..." is redundant. You could rephrase the first sentence and incorporate this sentence into it with a little effort.

-- Line 19: There have been actual in situ observations of relativistic electrons generated by shocks associated with foreshock transients like SLAMS [e.g., Masters et al., 2013; Wilson et al., 2016]. That is, your two references here are both simulation/theory papers when there has been actual observations of the phenomena you reference.

-- Lines 20-24: Numerous things have changed in foreshock and quasi-parallel shock research since 2005 [e.g., Burgess et al., 2012; Burgess and Scholer, 2013; Wilson, 2016].

=====
Typos/Grammar:

=====
Line 36: Use two ` instead of two ' to start quotes in LaTeX.

-- Line 39: "solar wind form a highly"  "solar wind forms a highly"

-- Line 41: "region create a complex region"  Perhaps find another word to avoid the awkward repetition here?

-- Line 48: "They have, e.g., been"  "They have been" or "They have been, e.g.,"

-- Lines 59-60: "...that for the first time allow to follow the jet formation at the shock..."  "that, for the first time, allow us to follow the jet formation at the shock"

-- Lines 76-77: "...to observe the evolution of both phenomena, originating at the upstream region, evolving and ending up..."  Again, it might be a good idea to avoid the repetition of the word evolve.

=====
References:
=====

-- Burgess, D., and M. Scholer "Microphysics of Quasi-parallel Shocks in Collisionless Plasmas," Space Sci. Rev. 178(2), pp. 513-533, doi:10.1007/s11214-013-9969-6, 2013.

-- Burgess, D., et al., "Ion Acceleration at the Earth's Bow Shock," Space Sci. Rev. 173, pp. 5--47, doi:10.1007/s11214-012-9901-5, 2012.

-- Chen, L.-J., et al., "Electron Bulk Acceleration and Thermalization at Earth's Quasiperpendicular Bow Shock," Phys. Rev. Lett. 120(22), pp. 225101, doi:10.1103/PhysRevLett.120.225101, 2018.

-- Lucek, E.A., et al., "Cluster magnetic field observations at a quasi-parallel bow shock," Ann. Geophys. 20, pp. 1699-1710, doi:10.5194/angeo-20-1699-2002, 2002.

-- Lucek, E.A., et al., "Cluster observations of structures at quasi-parallel bow shocks," Ann. Geophys. 22, pp. 2309-2313, doi:10.5194/angeo-22-2309-2004, 2004.

-- Lucek, E.A., et al., "Cluster observations of the Earth's quasi-parallel bow shock," J. Geophys. Res. 113, pp. 7, doi:10.1029/2007JA012756, 2008.

-- Masters, A., et al., "Electron acceleration to relativistic energies at a strong quasi-parallel shock wave," Nature Phys. 9, pp. 164-167, doi:10.1038/NPHYS2541, 2013.

-- Pollock, C.J., et al., "Fast Plasma Investigation for Magnetospheric Multiscale," Space Sci. Rev. 199(1), pp. 331--406, doi:10.1007/s11214-016-0245-4, 2016.

-- Stasiewicz, K., et al., "Properties of fast magnetosonic shocklets at the bow shock," Geophys. Res. Lett. 30(24), pp. 2241, doi:10.1029/2003GL017971, 2003.

-- Turner, D.L., et al., "Direct Multipoint Observations Capturing the Reformation of a Supercritical Fast Magnetosonic Shock," Astrophys. J. Lett. 911(2), pp. L31, doi:10.3847/2041-8213/abec78, 2021.

-- Wilson III, L.B., "Low frequency waves at and upstream of collisionless shocks," Geophys. Monogr. Ser. 216, pp. 269--291, In "Low-frequency Waves in Space Plasmas," Eds. A. Keiling, D.-H. Lee, & V. Nakariakov, American Geophysical Union, Washington, D.C., doi:10.1002/9781119055006.ch16, 2016.

-- Wilson III, L.B., et al., "Relativistic Electrons Produced by Foreshock Disturbances Observed Upstream of Earth's Bow Shock," *Phys. Rev. Lett.* 117(21), pp. 215101, doi:10.1103/PhysRevLett.117.215101, 2016.

Reply to the reviewer #1

Black = Reviewer's text/comment Blue = Author's reply
--

Summary:

The authors analyse data of a nicely spaced aligned configuration of the 4 MMS spacecraft covering the range of roughly 1000 km from upstream solar wind across the quasi-parallel bow shock to the downstream near shock magnetosheath. Roughly two spacecraft are in the highly variable upstream wave/shock region, and one shock downstream in the enhanced magnetic field and density magnetosheath. The upstream data show the typical signs of large amplitude upstream waves which are on the way to become the shock by steepening and reflecting particles back upstream which excite precursor whistlers. In order to get a causally connected evolution the spacecraft data have been Galilei shifted in time (this is useful when referring to observations in the magnetosheath where a so-called jet is detected). This evolution of the shock shows the typical signature of shock transitions noting that the front of the 3 shock crossings approaches the spacecraft and passes them over.

The magnetosheath spacecraft shows a structure different from the shock and normal sheath which is identified as the “jet” as a region of enhanced dynamical pressure, consisting of background sheath plasma and a stream-like ion population of nearly solar wind speed and temperature but higher density, i.e. compression. The required anticorrelation between magnetic field and density confirms this picture. The near solar wind speed of the “jet” indicates that it is a structure of the reformed quasi-parallel shock that has passed the shock, survived locally and moves downstream through the sheath. In this picture which the authors develop a sheath is a fraction of an upstream large amplitude foreshock wave packet which finds a dynamical hole to pass barely affected by the shock transition through the shock to the downstream region.

In looking at the solar wind speed (MMS1/2) it seems that the conditions were normal in the solar wind rather than to the slower wind speed.

Altogether this carefully done work on the sequence of events related to quasi-parallel collisionless shocks, and the detection of a process of how foreshock features can sometimes be transferred downstream as if no shock would have been present locally is interesting and worth publishing.

Thank you for your thorough review of our paper. We have replied to all of your points, which we think create an interesting discussion about the topic of magnetosheath jets. Below, we go through each of your comments in detail.

Comments:

I find the observations and the sequence intriguing and a nice contribution to the shock dynamics. It is clear that locally such a jet should affect the evolution of the sheath. The authors present a graphical model which represents their imagination of how the jet formation proceeds. This may or may not be the ultimate picture. To my taste Fig 4 a is not very helpful as the data already contain the physics in rather clearer form and the model does not clarify it in a better way. However this is a question of taste which may differ from mine.

We agree that it is a matter of taste but we think it can be a useful tool for discussing the interpretation of the data at the end of the manuscript. Therefore, we decided to keep it in the manuscript.

However, now we included some more information in the manuscript to make the connection between the observations and the modeling (changed to FIG. 6 in the latest manuscript) clearer, which we hope you, will find satisfactory.

The observation and identification of the jet and its division into the two populations and relation to shock reformation is most interesting. Obviously in the jet part the reformation has not been successful as it enabled part of the solar wind to pass across the shock, something which has already long been suspected that the shock ramp naturally contains holes where compressed and modified solar wind may pass through on some short tangential scale which is probably prescribed by the tangential wavelength of the large amplitude foreshock waves. Two adjacent of them definitely leave a region where the reformed shock is barely precisely defined.

We also believe that the partial moment derivation and the shock reformation are the most interesting observations. Specifically, currently we are conducting an ongoing work consisting of a comparison between jet observations for different plasma moment derivations. It appears that a few properties that are typically observed may be affected not only by resolution effect but as we show in this work, by the actual particle moment derivation.

The investigation and model is uni-dimensional prescribed by the configuration of the quartett of spacecraft. So nothing can be said about the interesting question of the tangential extension of the jet. It must have been tangentially local not distributed allover the surface of the bow shock. And during further propagation it should evolve, either steepening further into another local subshock or decaying by some kind of spatio-temporal dispersion. Magnetosheath flow should transport it sideways. When surviving up to the magnetopause its effect should locally cause a strong deformation of the latter.

We plan in the future to investigate through a combination of observations and computer simulation the question of the exact evolution a jet has as it propagates to the magnetopause. Furthermore, of special interest is to investigate the effect it may have to other phenomena such as magnetopause reconnection, connecting shock dynamics to Earth's inner magnetosphere effects. This is also planned to be investigated in the near future.

Altogether a nice work on which I do not have much criticism to add.

We once again thank the reviewer a lot for his kind words and positive view of our work.

Reply to the reviewer #2

*Note: All lines mentioned refer to the manuscript **with the tracked changes**.*

Black = Reviewer's text/comment
Blue = Author's reply

Summary:

=====
The paper argues that the four MMS spacecraft observed the bow shock reforming and generating supersonic magnetosheath jets in the process. While I fully agree that the evidence supports the interpretation of shock reformation, I am not sure about the observation of a magnetosheath jet. The supplemental figure does look to have a solar wind beam and some much hotter, more tenuous ions drifting relative to it. However, it is not clear this is indeed a jet and that the VDF was even observed downstream of the shock.

As a note, I will say that I am not opposed to the authors' interpretation nor am I opposed to seeing this manuscript again after revision. If the authors are correct in their interpretation, then this certainly would be an incredibly important paper for the physics community at large, not just the heliospheric and astrophysical communities. My main concerns are likely due to my misunderstanding, so I would like to see the authors responses before making any final decisions.

Thank you for your thorough review of our paper. We have addressed all of your points, which we think improved the overall quality of the paper. Below, we go through each of your comments in detail.

===== Figure related issues =====

A series of comments were made regarding the figures that we address all together, while also answered individually below. Specifically, the comments were:

- The label for panel (e) is missing. (Figure 1)
- The label for panels (d) and (e) are missing. (Figure 2)
- The label for MMS4 is missing above the blue triangle. (Figure 3)
- The text label for the purple region is missing. (Figure 4)

Upon investigating, we were unable to find these issues in our version of the manuscript. We tested if this was the result of printing or of different pdf viewer. It appears to be a “bug” with some .pdf viewers and heavy vector graphics. Specifically, the default viewer in mac OS appears to sometimes have this issue.

However, to avoid such technical issues in the future we changed all the figures from vector format to .png.

Regardless of that, all the figures have been re-created based on the editor's and the reviewers' suggestions. Different coloring has been used for spectrums (<https://www.nature.com/articles/s41467-020-19160-7>), larger font in both axis and labels have been added, and other minor improvements (e.g. mentioning of spacecraft number per figure) were also included.

Finally, figures 1 and 2 have been split into 2 different figures each that respectively show the measurements of the two “outermost” spacecraft and the two “innermost” ones. We believe that this enhances the readability of the manuscript and effectively makes the figures much larger and easier to view. Below we go point by point and reply to all the reviewer's comments.

General Comments:

-- Figure 1:

Figure 1 has been split into Figures 1 and 2 in the latest manuscript.

-- The label for panel (e) is missing.

Fixed

-- I am not convinced that the purple shaded region is a jet. The reason I say as much is for the following reasons:

The FPI DIS instrument is not really suited for measuring the solar wind ions [e.g., Pollock et al., 2016] and as the spacecraft rotates, and DIS scans, the instrument shows a periodicity in the intensity of the ion signal from the main solar wind core beam. If you examine the ion differential energy flux spectrogram and then the 1D reduced ion VDFs, you can see some of this periodicity. But more importantly, you can see that the main beam doesn't really change in the ion differential energy flux spectrogram during what the authors are calling a jet. Perhaps the color scale is saturated or I need new glasses, but I was under the impression that magnetosheath jets show a shift in energy of the peak and an enhancement in intensity of that shifted peak, relative to the surrounding plasma. There is a kind of one-sided shift in the location of the peak but it isn't very convincing. Figure 2 does a little better but I would suggest the authors find a way to make the contrast more obvious because I am really squinting to try and see this.

We investigated the problems that FPI instrument could have caused. After reading the recent work by [Roberts et al 2021] and examining the variations of our example we concluded that while indeed the FPI instrument is not ideal for solar wind ion related distributions, it is accurate enough for the purpose of this work. The variations of the velocity moments observed in FIGs 1,2 (b,c) and 2,3 (a,b) show measurements in the pristine solar wind to vary within $\sim \pm 10^0 - 10^1$ km/s. However, when analyzing the close to the shock downstream jet ($\sim 200 - 400$ km/s), we see that the variation observed may not cause any problem to the threshold we use to classify the jet observations (also discussed below).

Moving on to the most important aspect, we agree that although it was implied in the article, on the initial submission there is never explicitly mentioned what definition we use for the observed magnetosheath jet. We have updated the definition that we use for downstream jet in the methodology section at the end of the manuscript.

Line: 314-328

Specifically, we use the same definition as [Archer et al. 2013, Karlsson et al. 2015, Raptis et. al 2020, Palmroth et al. 2021] and other jet related articles, where a magnetosheath jet is defined as a dynamic pressure enhancement that exhibits at least a 100% enhancement in dynamic pressure compared to the surrounding magnetosheath.

However, in our example (previously FIG. 2, now FIGs. 3 and 4) we also include a background solar wind dynamic pressure value from OMNIweb in order to show that even with other type of criteria, the observed jet is above the threshold that are typically used for defining jets (relevant discussion of [Plaschke et al. 2018]). For example, [Dmitriev & Suvorova 2015] use a different criterion where they use both solar wind and magnetosheath measurements to define a jet. Our case would have satisfied such criteria as well.

With the now clearly written definition of a jet, we can see that the observed $\sim 200\%$ increase in dynamic pressure is well above the typically used threshold for magnetosheath jets.

Furthermore, due to our definition, the changes in the ion differential energy flux are not necessary. However, we agree that due to the velocity increase there will be a shift to higher energies. This however is indeed not easy to visualize sometimes. As a result, we have included in FIG. 4 (c) a zoomed in

version of the ion differential energy flux spectrogram. Indeed, as shown there, we do see a shift in energy, as expected by the reviewer.

Finally, as mentioned in lines 128-135, the properties observed by MMS3 are typical of downstream shocked plasma. While we definitely agree that it is hard to study structures close to the quasi-parallel shock environment, we have good evidence that the observed high velocity jet is indeed observed downstream of the shock for the following reasons:

- Density and magnetic field measurements have values that are typical of magnetosheath environment. This can be seen by comparing either from data directly from the same day (FIG. 2 (c) and FIG. 4(b), starting and ending time of MMS3) or from other studies (e.g. [Raptis et al. 2020]). This highlights the compression of the plasma as experienced through a shock crossing.
- In the ion energy spectrums, the 1D and the 2D reduced VDFs, we see a clear thermalized plasma population associated with the magnetosheath.
- Finally, the fact that there is no clear crossing after the observed jet (rather than typical values of the magnetosheath) shows that no back and forth movement of the shock could have made the observation to appear upstream of the shock. If such a motion was present, then either a clear shock transition would have been observed immediately before or after the jet observation, which is not the case in our example.

-- Figure 2:

Figure 2 has been split into Figures 3 and 4 in the latest manuscript.

-- The label for panels (d) and (e) are missing.

Fixed

-- Similar to my comment on Figure 1, could you find a way to make the contrast better? Perhaps show the Mach numbers of the partial velocity in panel (e) instead of the relevant speeds? I ask because it's difficult to see just how far above the magnetosonic speed the jet bulk speed lies. This is important as the uncertainty of the velocity moments can be very large even for instruments with much longer integration times. Since FPI DIS has such a short integration time, the signal-to-noise ratio is very small. This amplifies the uncertainty in the velocity moments. That its energy and angular resolution were designed for the magnetosheath only amplifies the issue in the solar wind. So if this jet is only ~5-10% above Mach 1, is that really statistically significant?

We agree that the scale of the velocity was not good enough for a quantitative comparison to be made. As a result, we have updated the FIG. 4(c) panel, which now includes the fast magnetosonic Mach number. The velocity of the jet reaches up to ~30-40% above Mach 1. This is significantly higher than the typical error that may occur in observations from FPI even in the case of solar wind measurements (<10% relatively to the velocities observed by the jet)

Furthermore, while the title and the manuscript mention the super-magnetosonic velocities, this is not a necessary property of jets either. The jet definition which is now included (lines: 314-328) implies that there are plenty of magnetosheath jets that do not exhibit super magnetosonic velocities. However, in our case we can see that our observations are well above Mach 1.

-- Figure 3:

In the new manuscript, FIG 3 is changed to FIG 5, due to the splitting of the first 2 figures.

-- The label for MMS4 is missing above the blue triangle.

As, answered above, now it should be visible.

-- Figure 4:

In the new manuscript, FIG 4 is changed to FIG 6, due to the splitting of the first 2 figures.

-- The text label for the purple region is missing.

Fixed

-- It would be very helpful, for readers like myself, if the MMS symbols also had numbers in this figure. Otherwise folks like myself with poor short-term memories keep scrolling back-and-forth with earlier pages.

We added the MMS numbers above the symbols.

-- This figure is difficult to disentangle. On the left, panel (a), the vertical axis is meant to imply time and the horizontal the spatial location along the shock normal, correct? Then I see eight vertical lines in panel (b) that are suppose to correspond to each of the horizontal color-coded bars in panel (a), right? I am confused because at t1 each spacecraft is seeing something very different. I also see the spacecraft symbols above panel (a) and vertical lines extending down. However, panel (a) and (b) doesn't seem to line up. For instance, the red diamond vertical line in panel (a) at t6 looks to be in the upstream solar wind. The vertical line in panel (b) at t6 for the red diamond looks to be in the shock ramp (you can see this from the deceleration of the main solar wind beam in the 1D reduced ion VDFs). Actually all four spacecraft look to be in a shock ramp at t6 according to the location of the t6 line in panel (b). I guess my point is that this figure seems inconsistent or I am completely misunderstanding it.

Indeed, there was an inconsistency with the measurements, which was fixed by changing the t6 time on panel (a) of the schematic (FIG. 6).

However, to avoid further confusion, apart from the change of t6, we have added an extra sentence reflecting the speculative aspect that is associated to the absent of *in-situ* measurements in most part of the schematic (see caption of FIG, 6). Even if the schematic is not 100% accurate on all details, we find it a very useful tool for discussing the interpretation of the data.

-- Assuming the purple region corresponds to the jet (inferred because jet is the only type of region missing), then is it the case that only MMS3 sees the jet? I think the authors implied as much in the previous sections, but the 1D reduced ion VDFs here do not really convey or illustrate this, so far as I can tell. I agree it looks like the authors are indeed observing shock reformation in one of its many manifestations, but I am having difficulty discerning why they argue there is a magnetosheath jet here.

We believe that with the discussion above, the new updated definition of jet in the methodology (lines 314-328) and the updated caption we have clarified this issue.

SLAMS:

-- Lines 89-93: Note that SLAMS exhibit linear, left-hand, and right-hand polarizations in the spacecraft frame [e.g., Wilson, 2016]. Thus, the left-hand polarization is not really conclusive. The enhancement in $|B|$ simultaneous with n is also not conclusive, as that is just a general property of waves on the fast/magnetosonic-whistler branch. Note that I am not arguing these fluctuations are not SLAMS, just pointing out that the evidence presented is not conclusive.

-- Note that SLAMS are always observed with diffuse ions [e.g., see description and examples in Wilson, 2016] and typically are distinguished from shocklets in that $dB/B > 2$.

-- Only a few of your red-shaded regions actually satisfy this latter criteria and it's not clear if any satisfy the former. The top-left MMS2 (and maybe top-right MMS3) example in Figure S1 kind of looks like diffuse ions, but it's impossible to say without 1D cuts through the VDFs (i.e., not the reduced VDFs). This is because the phase space density profile of diffuse ions is very hard, i.e., it can be nearly flat. My color fidelity is not sufficient to determine the velocity gradient from these 2D reduced ion VDFs.

We agree with the reviewer that not a fully detailed study of the observed structures (SLAMS or not) have been conducted. That was done to avoid focusing on that aspect, since the focus of this article is the under investigation jet.

However, this approach caused probably a larger confusion, so we have changed the word SLAMS to either *compressive structure* or to *compressive magnetic structure*. This was done to avoid diving into the differences between upstream structures such as shocklets and SLAMS, since in our work their exact distinction is not crucial.

Furthermore, we added references to recent works that have speculated on how foreshock structures (such as SLAMS) may be associated to the generation of jets (lines 58-62) [Karlsson et al. 2015, Raptis et al. 2020, Suni et al. 2021]. However, so far, none of these studies has provided any causal link or direct observations to such mechanism. Also, we changed the wording of the sentences related to the identification of structure “1” in lines 98-110 and in the methodology section.

Finally, we added lines (293-297) to illustrate the importance of continuing the research towards understanding the exact properties of the observed compressive structures.

-- Even if the top-left MMS2 example in Figure S1 is representative of diffuse ions, is this ion VDF seen simultaneously with what the authors are calling SLAMS? The example red-shaded regions in Figure 1e do not, I think, satisfy the $\text{dB/B} > 2$ criteria.

Yes, the VDF shown in the supplementary material is taken during the same time as the red-shaded region (1).

-- The scale size of SLAMS is only ~ 1000 km along the normal to their direction of propagation and only slightly larger along the transverse [Lucek et al., 2002, 2004, 2008]. The lateral separation between MMS2 and MMS3 is over 800 km, which is coming close to the relevant scale size of SLAMS. So if these oscillations are SLAMS, is it really surprising that you see such large differences in the magnetic field between the different spacecraft? You can see that MMS1 and MMS2 are the most similar in the magnetic field profiles, and these spacecraft are spatially adjacent to each other. The profile starts to degrade at the next adjacent spacecraft, MMS4, then "falls apart" at the furthest spacecraft, MMS3.

Indeed as the reviewer writes, SLAMS have scale sizes typically ~ 1000 km long and the separation is very similar to that. This is a strong advantage of the presented event since it allows us to study the evolution of these structures. We have added references of [Lucek et al., 2004, 2008, Wilson, 2016] at this point referring to the basic properties of SLAMS in lines: 103-110.

In addition, indeed we agree that the observed differences are not surprising (similar to what we see in [Liu et al. 2021] and [Turner et al. 2021]), since these structure have their own evolution cycle and part of the observed differences appear to be also a result of that evolution (e.g. shown in FIG. 5)

-- Lines 172-188:

-- Turner et al. [2021] did not discuss gyrotrapping, so far as I know. They did observe a phase shift between the peak $|\text{dB}|$ enhancement and the peak dn enhancement, but this has already been explained as a consequence of nonlinearity [e.g., Stasiewicz et al., 2003].

-- Generally, $|\text{dB}|$ and dn oscillate in phase within magnetosonic whistler precursors [e.g., Chen et al., 2018; Turner et al., 2021], which is what I am seeing in your Figure 2.

Various changes and additions have been made to reflect the updated information.

In our case, we do observe a slight phase shift similar to figure 2 of Turner et al. [2021], which is now more carefully elaborated in the manuscript. We also separated the gyrotrapping discussed by Chen et al 2021 in a separate sentence to avoid any confusion with the references.

Lines: 191-200

-- Lines 198-208:

-- Things that starts upstream and transmit into the downstream are generally compressed, not expanded, correct? Why is the pile-up region expanded in the downstream?

There appears to be a confusion regarding the definition of the pile-up region. As a result, we re-wrote the following parts that mention the pile-up region:

By pile up region we refer to the part of the evolved whistler precursor waves that is part of the evolution of the structure (#1). As a result, we don't mean that an expansion took place when the compressive structure (#1) moved downstream. We mean that if now one adds the "pile-up" region, to the structure, from an observational perspective, the whole area of enhanced magnetic field and density is extended.

Relevant changes in manuscript:

Lines: 154

Lines: 227 – 231

Lines: 234 – 239

-- How does an oscillation lose its polarization? This would require a mode conversion or some sort of wave-wave interaction or some type of destructive or constructive interference, would it not? That is, you cannot arbitrarily remove angular momentum from the electromagnetic field. So how is it happening here?

As a follow up to our previous point, again we do not mean that the initial structure (#1) is not polarized anymore after crossing downstream of the shock. On the contrary, by looking at FIG 5. We can see that even downstream, the structure itself appears to still have a very similar polarization signature.

What we want to focus on with this discussion is that if we only had seen this whole region from the point of view of MMS3, we would not be able to distinguish the different structures that exist there (SLAMS and evolved whistler waves). To avoid any confusion we removed the phrase regarding polarization and we focus on the fact that the origin of the structure would have been unknown if it was not for the MMS turbulence campaign providing measurements both upstream and downstream of the shock.

Lines: 154

Lines: 227 – 231

Lines: 234 – 239

Minor Concerns:

-- Lines 14-16: The sentence starting with "The Earth's bow shock..." is redundant. You could rephrase the first sentence and incorporate this sentence into it with a little effort.

Changed.

-- Line 19: There have been actual in situ observations of relativistic electrons generated by shocks associated with foreshock transients like SLAMS [e.g., Masters et al., 2013; Wilson et al., 2016]. That is, your two references here are both simulation/theory papers when there has been actual observations of the phenomena you reference.

The references have been changed to include more recent work and direct observations that have been made.

-- Lines 20-24: Numerous things have changed in foreshock and quasiparallel shock research since 2005 [e.g., Burgess et al., 2012; Burgess and Scholer, 2013; Wilson, 2016].

The references have been changed to refer to more recent work.

Typos/Grammar:

-- Line 36: Use two ` instead of two ' to start quotes in LaTeX.

Changed throughout the manuscript.

-- Line 39: "solar wind form a highly"  "solar wind forms a highly"

Fixed.

-- Line 41: "region create a complex region"  Perhaps find another word to avoid the awkward repetition here?

Changed 2nd region to environment.

-- Line 48: "They have, e.g., been"  "They have been" or "They have been, e.g.,"

Fixed.

-- Lines 59-60: "...that for the first time allow to follow the jet formation at the shock..."  "that, for the first time, allow us to follow the jet formation at the shock"

Fixed

-- Lines 76-77: "...to observe the evolution of both phenomena, originating at the upstream region, evolving and ending up..."  Again, it might be a good idea to avoid the repetition of the word evolve.

Change the word evolution to development.

References

Archer, M. O., & Horbury, T. S. (2013). Magnetosheath dynamic pressure enhancements: Occurrence and typical properties. *Annales Geophysicae*, 31, 319.

Dmitriev, A. V., and A. V. Suvorova. "Large-scale jets in the magnetosheath and plasma penetration across the magnetopause: THEMIS observations." *Journal of Geophysical Research: Space Physics* 120.6 (2015): 4423-4437.

Karlsson, T., Kullen, A., Liljeblad, E., Brenning, N., Nilsson, H., Gunell, H., & Hamrin, M. (2015). On the origin of magnetosheath plasmoids and their relation to magnetosheath jets. *Journal of Geophysical Research: Space Physics*, 120, 7390–7403.

Liu, T. Z., Hao, Y., Wilson, L. B., Turner, D. L., & Zhang, H. (2021). Magnetospheric multiscale observations of Earth's oblique bow shock reformation by foreshock ultralow-frequency waves. *Geophysical Research Letters*, 47, e2020GL091184. <https://doi.org/10.1029/2020GL091184>

Lucek, E.A., et al., "Cluster observations of structures at quasiparallel bow shocks," *Ann. Geophys.* 22, pp. 2309-2313, doi:10.5194/angeo-22-2309-2004, 2004.

Lucek, E.A., et al., "Cluster observations of the Earth's quasiparallel bow shock," *J. Geophys. Res.* 113, pp. 7, doi:10.1029/2007JA012756, 2008.

Palmroth, Minna, et al. "Magnetosheath jet evolution as a function of lifetime: global hybrid-Vlasov simulations compared to MMS observations." *Annales Geophysicae*. Vol. 39. No. 2. Copernicus GmbH, 2021.

Plaschke, F., Hietala, H., Archer, M., Blanco-Cano, X., Kajdič, P., Karlsson, T., Lee, S. H., Omid, N., Palmroth, M., Roytershteyn, V., Schmid, D., Sergeev, V., & Sibeck, D. (2018). Jets downstream of collisionless shocks. *Space Science Reviews*, 214(5), 81.

Raptis, Savvas, et al. "Classifying magnetosheath jets using MMS: Statistical properties." *Journal of Geophysical Research: Space Physics* 125.11 (2020): e2019JA027754.

Roberts, O. W., Nakamura, R., Coffey, V. N., Gershman, D. J., Volwerk, M., Varsani, A., et al. (2021). A Study of the Solar Wind Ion and Electron Measurements from the Magnetospheric Multiscale Mission's Fast Plasma Investigation. *Journal of Geophysical Research: Space Physics*, 126, e2021JA029784. <https://doi.org/10.1029/2021JA029784>

Suni, J., Palmroth, M., Turc, L., Battarbee, M., Johlander, A., Tarvus, V., et al. (2021). Connection between foreshock structures and the generation of magnetosheath jets: Vlasiator results. *Geophysical Research Letters*, 48, e2021GL095655. <https://doi.org/10.1029/2021GL095655>

Wilson III, L.B., "Low frequency waves at and upstream of collisionless shocks," *Geophys. Monogr. Ser.* 216, pp. 269–291, In "Low-frequency Waves in Space Plasmas," Eds. A. Keiling, D.-H. Lee, & V. Nakariakov, American Geophysical Union, Washington, D.C., doi:10.1002/9781119055006.ch16, 2016.

REVIEWER COMMENTS

Reviewer #1 (Remarks to the Author):

I am impressed by the detailed answers to the comments of both us referees.

The authors have defended almost all their points at least mostly to my satisfaction. Though I nevertheless am not completely convinced that the structure they see is indeed what can be doubtlessly called a supermagnetosonic jet and no remainder from the quasi-parallel shock, it is my strong feeling that the paper is fine and should be published. The sequence of data is really unique and impressive and should be presented to the audience.

I therefore accept the paper for publication, with one very little change I would like to see:

Please erase the word "clear" before evidence in line 68. No doubt, the paper presents evidence, but clear evidence is a bit strong as the proof (the detailed comments of the referee shows that the evidence is not completely clear, and the detailed answers do not make it completely clear) is still open to maybe slightly different interpretations only.

Otherwise the paper is ok. That the authors insist on their Fig 6a is, to my opinion, not advantageous to the paper, but that it is in the responsibility of the authors to keep it or not. I will not insist on its deletion. To me it does not say much. The data instead are really wonderful. But as always with data, one can have different opinions about them.

This does not affect publication, however.

Reviewer #2 (Remarks to the Author):

The authors have addressed all of my concerns. I see no reason to further delay publication.

I will note that there is a very big difference between statistical uncertainties and physical uncertainties in particle detectors. The uncertainties reported by the FPI team in their datasets are statistical uncertainties based on idealistic assumptions. While the FPI data is by far the best to date of any space plasma investigation ever flown, it is not as accurate as many believe. That is, the velocity moments in the solar wind are rarely going to be accurate to better than 10%. The work by Roberts et al. [2021] uses averages, among other things, to compare intervals between OMNI and MMS. While the bulk flow velocities are generally in the right ballpark, the density and temperatures are not. That is, you may find the bulk flow velocities hovering near the 10% mark but the number densities and temperatures will be much more problematic. This is not a criticism of FPI because FPI was not designed for the solar wind. It's more a word of caution as the author's response seems to imply more confidence in the instrument's capabilities than I think is warranted. Again, this is just a very nit-picky nuance I wanted to clarify, not a criticism of the paper or the instrument.

Reply to the reviewer #1

Black = Reviewer's text/comment Blue = Author's reply
--

I am impressed by the detailed answers to the comments of both us referees.

We thank you a lot for your kind words.

The authors have defended almost all their points at least mostly to my satisfaction. Though I nevertheless am not completely convinced that the structure they see is indeed what can be doubtlessly called a supermagnetosonic jet and no remainder from the quasi-parallel shock, it is my strong feeling that the paper is fine and should be published. The sequence of data is really unique and impressive and should be presented to the audience.

I therefore accept the paper for publication, with one very little change I would like to see:

Please erase the word “clear“ before evidence in line 68. No doubt, the paper presents evidence, but clear evidence is a bit strong as the proof (the detailed comments of the referee shows that the evidence is not completely clear, and the detailed answers do not make it completely clear) is still open to maybe slightly different interpretations only.

We agree with the reviewer and we have removed the word “clear” to show that measurements are open to different interpretations and discussion. Furthermore, we added a discussion about the possible uncertainties that are present in such in-situ observations (lines 317-324).

Otherwise the paper is ok. That the authors insist on their Fig 6a is, to my opinion, not advantageous to the paper, but that it is in the responsibility of the authors to keep it or not. I will not insist on its deletion. To me it does not say much. The data instead are really wonderful. But as always with data, one can have different opinions about them.

While we agree about the wonderful data that MMS provided us to analyze, we feel that for the non-trained readers of such a broad audience journal, FIG 6a may serve a purpose. Even if to some readers, their ability to visualize the phenomenon makes it redundant, we believe that it makes the result of our work accessible to a larger audience (e.g. those that have not worked with in-situ measurements). We however keep your comments for the later studies we are planning to do.

This does not affect publication, however.

Once again, we thank you for your thorough review and your insightful comments.

Reply to the reviewer #2

Black = Reviewer's text/comment Blue = Author's reply
--

The authors have addressed all of my concerns. I see no reason to further delay publication.

We are very glad that you are satisfied with all our replies and we thank you a lot for your contribution in making the manuscript significantly better through your comments.

I will note that there is a very big difference between statistical uncertainties and physical uncertainties in particle detectors. The uncertainties reported by the FPI team in their datasets are statistical uncertainties based on idealistic assumptions. While the FPI data is by far the best to date of any space plasma investigation ever flown, it is not as accurate as many believe. That is, the velocity moments in the solar wind are rarely going to be accurate to better than 10%. The work by Roberts et al. [2021] uses averages, among other things, to compare intervals between OMNI and MMS. While the bulk flow velocities are generally in the right ballpark, the density and temperatures are not. That is, you may find the bulk flow velocities hovering near the 10% mark but the number densities and temperatures will be much more problematic. This is not a criticism of FPI because FPI was not designed for the solar wind. It's more a word of caution as the author's response seems to imply more confidence in the instrument's capabilities than I think is warranted. Again, this is just a very nit-picky nuance I wanted to clarify, not a criticism of the paper or the instrument.

We agree with the reviewer and we would like to clarify that it was never our intention to appear overconfidence regarding FPI usage. As the reviewer says, the moment derivation of the FPI under such conditions may as well provide uncertainties, not only statistically but also physically, due to the particle detectors used.

We also thank the reviewer for their reminder. In future planned work that uses FPI to study plasma moments such as number density and temperature variations, we will most certainly approach it with caution.

Finally, we added a short discussion regarding the uncertainties of FPI measurements in the solar wind at the methodology sections (lines 317-324) reflecting the discussion that took place during the revision process.